# African Swine Fever in the Philippines: A Review on Surveillance, Prevention, and Control Strategies

**DOI:** 10.3390/ani14121816

**Published:** 2024-06-18

**Authors:** Cherry P. Fernandez-Colorado, Woo Hyun Kim, Rochelle A. Flores, Wongi Min

**Affiliations:** 1Department of Veterinary Paraclinical Sciences, College of Veterinary Medicine, University of the Philippines Los Baños, Los Baños 4031, Laguna, Philippines; 2College of Veterinary Medicine & Institute of Animal Medicine, Gyeongsang National University, Jinju 52828, Republic of Korea; woohyun.kim@gnu.ac.kr (W.H.K.); floresrochellea@gmail.com (R.A.F.); wongimin@gnu.ac.kr (W.M.)

**Keywords:** African swine fever, epidemiology, surveillance, control strategy, Philippines

## Abstract

**Simple Summary:**

ASF remains a significant concern for the Philippines as it has impacted the swine industry and food security. It causes high mortality rates, leading to decreased pork production, high prices of pork and pork products, and financial losses for farmers and stakeholders, especially for the livelihoods of many small-scale pig farmers in the Philippines. Since its first appearance in 2019, the government has implemented strict measures to control the spread of the disease that include enhanced surveillance, strict biosecurity and quarantine protocols, restricted movement of live pigs and pork products, and culling of infected and at-risk pigs. Ongoing surveillance and cooperation between industry stakeholders, government agencies, and research institutions and collaboration with international organizations and experts to develop enhanced strategies for ASF control and prevention are necessary to mitigate ASF’s impacts on the swine industry. Moreover, public awareness campaigns and support for affected farmers are continuously in place to combat ASF in the Philippines.

**Abstract:**

African swine fever (ASF), a highly contagious disease of swine, has posed a significant global threat to the swine industry. As an archipelago, the Philippines has a geographic advantage when it comes to the risk of ASF transmission. However, since its introduction to the Philippines in 2019, it has proliferated not only in backyard and commercial farms but also in wild pig populations. While certain parts of the country were more affected than others, the epidemiologic features of ASF necessitate that all affected areas must be closely monitored and that confirmed cases be treated with the utmost care. With the very limited data on ASF epidemiology and surveillance in the Philippines, future efforts to combat ASF must place even greater emphasis on improved prevention and control strategies. It is worth mentioning that the government’s efforts toward comprehensive ASF surveillance and epidemiological investigation into the possible ASFV sources or transmission pathways are the most important measures in the prevention and control of ASF outbreaks. This review article provides a comprehensive overview of the current swine industry and ASF situation in the Philippines, which includes its epidemiology, surveillance, prevention, and control strategies.

## 1. Introduction

Both domestic and wild species of pigs, except some natural hosts like warthogs and bush pigs, are susceptible to African swine fever (ASF), a non-zoonotic viral infectious hemorrhagic disease caused by a large, complex, and multi-enveloped DNA arbovirus from the genus Asfivirus, family Asfarviridae, with a viral genome length of 170–192 kb [1,2,3,4,5]. ASF is viewed as a serious threat to the swine industry wherever it is found and is classified as a notifiable transboundary disease due to its high virulence, high mortality rate, and very high socio-economic impact [2,6,7]. The severity of ASF outbreaks is demonstrated by the abrupt death of pigs within 4 days of infection and a 100% mortality rate in acute and peracute cases. Pigs infected with ASFV frequently exhibit non-specific clinical signs, such as fever, thrombocytopenia, skin lesions, loss of appetite, diarrhea, and vomiting, as well as hemorrhagic-related symptoms like vasculitis, skin erythema, and pulmonary edema [8,9,10]. The ASFV genome is very large and complex, which contributes to the difficulty of developing an effective vaccine. However, extensive efforts around the world have shown promising progress directed toward effective vaccine development. Various approaches have been employed, which include inactivated, subunit, DNA, virus-vectored, and live-attenuated (LAV) ASF vaccines [11]. Among these approaches, inactivated and subunit vaccines are safe but do not induce protective immunity [12,13,14] while live-attenuated (LAV) ASF vaccines are the most promising and exhibit a wide range of safety and efficacy against ASF [15]. Although it poses no threat to humans, ASF outbreaks can have a major effect on the pork sector, resulting in national and international restrictions on trade, affecting food security, and leading to major financial economic losses.

Originally endemic to Sub-Saharan Africa, ASF has spread globally to several Asian and European regions since it was first confirmed and reported in Georgia in 2007 [16,17]. In 2018, ASF p72 genotype II was found in China [4]. Given that nearly half of the world’s pigs are produced in China, its emergence in China has raised concerns for the swine industry both in Asia and globally. Subsequently, the disease has spread throughout numerous Southeast Asian nations between 2019 and 2023, including the Philippines, Vietnam, Laos, Cambodia, Timor-Leste, Myanmar, Singapore, Malaysia, and Indonesia [10] with ASFV genotype II as the prevailing strain. Currently, there are 24 known ASFV genotypes, however, significant new information on the genetic features of ASFV in relation to changes in virulence and genetic recombination, has been reported in China [18]. In this study, three recombinants of genotypes I and II were detected in pigs, and based on the B646L gene, these recombinants are categorized as genotype I due to their genetic similarity. However, 56% of their genomes are derived from genotype II ASFV. Furthermore, one of the recombinant viruses has been shown to be highly lethal and transmissible in pigs in animal experiments [18].

As of March 2024, the majority of the Philippines’ swine population falls under smallhold farms (70.4%), while commercial farms hold 27.1%, and semi-commercial farms hold 2.4% [19]. The total swine inventory of the Philippines in 2019 prior to the ASF outbreaks (as of July 2019) was estimated at 12.70 million heads as compared to the registered swine population in the same period of 2020, which only reached 11.74 million heads with a decline of roughly 18.6% [20,21]. The total swine inventory was estimated at 9.86 million heads as of September 2023, which is 2.1% lower than the 2022 total swine inventory of 10.07 million heads in the same period [22]. Since its detection in 2019, 89% (73 out of 82) of the provinces were already affected with 5 million pigs already killed according to the Pork Producers Federation of the Philippines which resulted in approximately PHP 200 billion or more losses [23].

The first ASF case in the Philippines was confirmed in Rizal province in July 2019, and since then ASF has quickly spread throughout the country, resulting in widespread outbreaks in all 17 administrative regions, with a total of 73 affected provinces as of April 2024, leading to substantial financial losses for both individual pig producers and the swine industry [9,24,25]. The introduction of ASF into the Philippines has caused significant economic setbacks in international animal importation and trade as well as in many impoverished households that rely on pigs to pay for many life necessities. This review aims to understand the current status of ASF and to suggest the key elements that affect the transmission, diagnosis, control, and prevention of ASF in the Philippines.

## 2. Socio-Economic Impacts of ASF

The swine industry has suffered significant economic setbacks as a result of the introduction of ASF into the country. Despite the continued threat of ASF outbreaks, the socio-economic impacts of ASF spread in the country need to be emphasized. In 2018, the total swine population was 12.71 million heads, which was a 0.83% increase from the 2017 inventory [20]. Since the disease was first reported in July 2019, the swine industry has dropped significantly in terms of production and inventory in the succeeding years (Figure 1A–C). Consequently, this led to a domino effect of initial drops in the value of production and farmgate price in the last quarter of 2019 until mid-2020, when the north and south areas of Luzon showed 16% and 12% reductions, respectively [26]. Thereafter, an increase both in cost of production and farmgate price followed (Figure 1B,C), ultimately affecting both farmers and consumers. This drop was attributed to a reduced pork demand due to the “ASF-scare” leading to a drop in sales of pork meat in retail stores, with losses of approximately PHP 50 billion (USD 894 million) [27]. Consequently, the sharp rise in pork prices has had a knock-on effect, driving up the cost of other meats, vegetables, and staple foods, particularly chicken, which may have served as an alternative source of protein. In terms of swine production by region, the bulk of the swine inventory distribution comes from Central Luzon (Region III) and CALABARZON (Region IV-A). However, the first incidence reports of ASF emerged from the Provinces of Rizal (Rodriquez and Antipolo) and Bulacan (Guiguinto), which are part of the CALABARZON and Central Luzon regions, respectively (Figure 2A) [28]. Consequentially, a sharp decline in the inventory of pigs was observed in these regions compared to other regions during the succeeding years (Figure 2B). According to Cooper et al., the ASF outbreak caused a domino effect because the negative income of backyard farmers meant that they were not able to pay their debts to feed suppliers [25]. Aside from this, watching the slaughtering of infected pigs as part of the protocol to control the outbreaks caused the farmers emotional trauma, stress, and depression. Moreover, veterinary medicine products also showed a reduced sales volume due to hog population loss [26]. Repopulation attempts in swine farms also faced difficulty in implementation due to the lack of strict biosecurity measures in farms and an increase in disinfectants used for sanitation or disinfection prior to repopulation [29]. As such, pork shortages may still be inevitable until unified science-based guidelines and an effective ASF vaccine become available for ASF control.

## 3. ASF Epidemiology and Surveillance

### 3.1. Epidemiology of ASF in the Philippines

Since its first discovery in the 1920s in Kenya, East Africa, ASF has affected other countries across Africa and consequently emerged in Europe and South America in the 1960s [30,31,32]. In 2007, it was reported to have infected the wild boar populations in Georgia [33]. Transmission continued to neighboring countries east of Georgia, particularly in Russia, until outbreaks were reported in China, specifically in Northeast China, in August 2018. It caused a substantial impact on the swine industry as China accounts for 50% of the global pig population [4,32,34,35].

Less than a year after entering China, the Philippines reported its first ASF outbreak in a backyard farm in Rizal province in July 2019, resulting in a 9.8% decline in pig production in the last quarter of 2019 [10,25,36,37]. According to the retrospective study on the epidemiology of ASF outbreaks in the Philippines, the first case of ASF in Rodriguez, Rizal was attributed to swill feeding practices of hog raisers in the area [38]. The question of how the disease may have been introduced into the Philippines remains unanswered due to a lack of concrete evidence and only causal inference as to the source of infection [39]. However, in June 2019, prior to the detection of ASFV in pigs, it was reported that luncheon meat seized at an international airport was positive for ASFV as confirmed by PCR performed at a Regional Animal Disease Diagnostic Laboratory (RADDL) [40]. Since then, ASFV occurrence in the country started with seven outbreaks, and in the span of approximately 2 months, ASF had spread to 31 provinces located in eight different regions in Luzon [25]. A recent study evaluated the most important factors influencing the spread, diagnosis, and control of ASF in the Philippines. Based on the risk factor analysis conducted, swill feeding, inadequate biosecurity protocols, and movement of personnel were identified as the risk factors that exacerbate the transmission of ASF among farms in the country [10].

To better understand outbreak trends and map the geographic locations of ASF viral genotypes or strains of concern, molecular characterization, full-genome sequencing and other epidemiological studies of ASFV local isolates are essential. Molecular epidemiology is a powerful tool that can be utilized to trace viruses during outbreaks, potentially leading to the identification of novel variants that could alter the phenotype of the virus. Although very limited, efforts have been made to understand the epidemiological occurrence of ASF in the Philippines (Figure 3). In 2020, the partial sequence of vp72 of ASFV showed that the ASFV strain in some samples originated from an outbreak in a province where 97–100% strains were closely related to genotype II ASFV from Georgia, Russia, Estonia, Poland, Belgium, China, and Vietnam [41]. Another study also revealed that the ASFV circulating in some affected areas of Central Luzon showed 100% sequence similarity with sequences reported from India, Malaysia, and Vietnam [42]. Recently, the coding-complete genome sequence of ASFV, a single contig with 192,377 bp isolated from an outbreak in 2021 and locally sequenced using Nanopore sequencing technology, was first reported and showed that the local isolate belongs to the same clade as ASFV strains from Timor-Leste, China, Ukraine, Belgium, and Poland [43]. These sequence data came from the blood samples of pigs from ASF outbreaks in two different provinces. Although very limited sequence data are available, this initial effort to sequence the local ASFV isolate is a good start. There is still a need to isolate and sequence more ASFV strains from different samples, hosts, and areas in the country. Full-length genomic sequencing of field isolates from ASF outbreaks in the country is currently ongoing. As of writing this review, 20 Philippine ASFV isolate whole genome sequences are available in NCBI GenBank; however, reports on these isolates’ genetic characterization and analysis including genetic variation still need to be elucidated and are beyond the scope of this review.

Virus isolation is still a challenge in the Philippines, as it requires an established laboratory with biosecurity practices and procedures capability as well as an optimized protocol to perform cell cultures and obtain virus stocks for use in whole-genome characterization and other future studies, such as host–pathogen interactions, viral pathogenesis, and local vaccine development. Up-to-date, acceptable quality data on the presence of ASF are needed to understand the epidemiological situation of the country. This information will advance our understanding of disease epidemiology and help to develop sensible public health recommendations for disease surveillance, monitoring, and control as well as information sharing with the general public.

With its potential for extensive and rapid geographical spread, it is imperative to develop effective control strategies by understanding ASF dynamics and identifying potential risk factors that may contribute to its transmission [44]. A comprehensive evaluation of the temporal and spatial dynamics of ASF in the Philippines was also conducted to understand the spread patterns of ASFV [10]. The results showed that there was a seasonal pattern of ASF occurrences from August to October with high frequencies, while low frequencies were observed from April to May. In addition, ASF cases during the second half of the year were found to be more severe in Luzon [10]. Hence, it is essential for the government to consider stricter implementations of its control and prevention strategies during the said season to reduce the potential impact of ASF outbreaks.

### 3.2. Surveillance and Diagnosis of ASF in the Philippines

Prevention through early detection and enhanced biosecurity play a very crucial role in ASF control. As in other countries, the Philippines also implemented surveillance programs for domestic pigs. This was performed early on, but routine surveillance was limited to the collection and testing of blood samples from live pigs in commercial and backyard farms only [25]. The most effective measure for controlling the virus is an appropriate surveillance program that is capable of early detection of the disease in both domestic and wild populations leading to the implementation of consolidated contingency plans upon detection of the virus.

The Philippines currently utilizes blood samples for ASF surveillance employing PCR-based testing systems managed by accredited government and private diagnostic laboratories (Figure 4). Although these PCR-based methods are specific and sensitive for ASF diagnosis, they are costly and laborious as blood collection alone requires pig handling for proper physical restraint. Moreover, environmental contamination at the collection site may result in an unintentional spread of the virus. In response to the need for point-of-care testing to address the long turnaround of results using PCR methods during ongoing ASF outbreaks, the Department of Agriculture has financially supported the development of an ASFV Nanogold Biosensor test kit, a nucleic acid-based assay that works in combination with LAMP and nanotechnology. Gold nanoparticles are utilized to interact with the amplified DNA (Figure 4) [45]. This test kit was validated in 41 farms (32 commercial, 9 backyard) in 7 provinces with 90–100% sensitivity and 77–85.7% specificity in parallel with RT-PCR [46]. Furthermore, it can be used in various samples such as oral and rectal swabs, feces, water, semen, feeds, blood, environmental swabs, and raw and processed meat samples to detect the P72 gene of ASFV. It can be used as a screening test which is on standby at the port of entry for meat products entering into the country [47].

Data on ASF surveillance are limited to ASFV-positive blood samples collected from different provinces in the country as a result of the government’s routine monitoring and during ASF outbreaks. These are reflected and updated monthly in the zoning map (Figure 5) and in areas with current ASF outbreaks (Figure 6) as reported by the National ASF Prevention and Control Programme (NASFPCP) of the Department of Agriculture-Bureau of Animal Industry (DA-BAI) [24].

The archipelagic nature of the Philippines poses a difficult challenge to the surveillance of ASF and even other diseases [48]. ASF outbreak management and eradication are difficult, costly, and resource-intensive due to ASF’s epidemiological characteristics such as many transmission pathways and reservoir populations [16]. Thus, systematic and efficacious methods for discerning where or what aspects to focus on for comprehensive surveillance of ASF are beneficial while reducing the negative economic impact of ASF outbreaks in the Philippines. Considerable efforts toward epidemiological surveillance of ASF have been the focus of several studies, not only on live domestic pigs but also on fresh or frozen meat and processed pork products (Figure 4). Despite imposed regulations and import taxes designed in each country to control the entry of pork and processed products, illegal importation remains a significant concern in the control and prevention of ASF outbreaks. In Taiwan, some pork products such as grilled pork, ham, and sausages confiscated at their airports tested positive for ASFV, and it is believed that ASF was introduced to the country through these contaminated pork products [49]. On the other hand, Japan also evaluated the risk of ASFV transmission from pork products that brought by tourists. The findings indicated that the likelihood of ASFV entering Japan increases substantially as more of these pork products are being fed to pigs in Japan [50]. Moreover, South Korea has conducting surveillance on pork products coming from countries affected by ASF since 2015, and in Thailand, ASFV has been detected in raw and processed pork products seized from tourists at international airports and quarantine checkpoints [51,52]. Neighboring countries in Asia have been collectively conducting surveillance on pork and pork products as a countermeasure to the ASF outbreak. Based on these studies, raw and processed pork products may pose a high risk of ASF transmission. The risk of ASF transmission among domestic pigs into other areas that are still ASF-free may be intensified with ASFV-contaminated raw meat and processed pork products sourced locally and imported from other countries. By imposing strict sanitary and import restrictions on raw meat and processed products, the entry of other ASFV variants or recombinants can be prevented, thus minimizing the impact of new outbreaks. These contaminated pork products are currently the main risk of ASF transmission in the country. While the Philippines is aware of the ability of ASFV to infect and persist in raw pork and pork products, initial detection of the virus was conducted in pork products from public markets in Metro Manila [53]. However, routine surveillance through molecular detection and transmission risk analysis of ASFV in raw meat and processed pork products on a larger scale is essential and still ongoing. Identifying the presence of ASFV in processed pork products in the Philippines would provide additional understanding of the complex epidemiology of ASF, as these products are transported across different regions that are possibly disease-free.

Vergerne et al. have discussed the high risk of ASF spillover from domestic pigs to wild populations, making it highly probable that the virus is circulating freely among wild pig populations in mainland Asia, particularly where habitats are contiguous to China [54]. The authors reported that there was a much lower ratio of ASF in wild boars to farm outbreaks in Asia as compared to Europe. Aside from South Korea with 605 infected wild boar carcasses since 2020, only China and Laos have reported wild boar infections with 23 cases [54]. Cases of ASF in wild pigs have been reported in the country. Following the deaths of nine Philippine warty pigs (*Sus philippensis*) in a privately owned forest patch in Davao del Norte, Mindanao, in January 2021, a postmortem examination and a PCR assay were conducted, confirming that ASFV was the cause of death. Moreover, two wild pigs exhibited clinical signs similar to those seen in ASFV positive domestic pigs [55]. Pig hunters reported approximately 100 wild pig deaths in Abra province in May 2021, revealing a positive test in meat samples [56,57]. Recently, a study showed that five out of fifteen fecal samples from wild pigs in a conservation center tested positive for ASFV [58]. There are no reports of dead Visayan warty pigs (*S. cebifrons*) in the wild as of March 2023, indicating that the ASF-positive wild pigs from the conservation center are still alive two years after the sample was collected in February 2021. Because backyard and small-hold farms in the country are set up with very little consideration of biosecurity, it is very likely that spillover events occur in wild populations much faster than the country can control. Therefore, it is necessary to also include wild pigs in the epidemiological model as a safety net not just for endemic, wild, and critically endangered species but also to provide decision-makers with a complete picture to utilize in setting policies on ASF control and mitigation.

Several epidemiological studies have demonstrated the ease with which ASFV can spread through contaminated bedding, clothing, shoes, farm environments, equipment, food waste, and other invertebrate vectors, either directly or indirectly [9]. According to reports, ASFV can survive for 30 days at room temperature in contaminated feeds and for 60 days in contaminated water. The ASFV genome has also been found in the larvae of house flies (*Musca domestica*), lice (*Haematopinus suis*), mosquitoes (*Culex pipiens*), stable flies (*Stomoxys calcitrans*), assassin bugs (*Triatoma gerstaeckeri*), hard ticks (*Ixodes ricinus*, *Amblyomma* spp., and *Dermacentor reticulatus*), and horse flies (*Tabanus* spp.) [5,9]. In the Philippines, swine farms frequently house a variety of invertebrates, pests, wild birds, and other animals. However, no research has investigated the possible roles that these animals may play in the spread of ASFV.

## 4. ASF Control and Prevention Strategies

ASF has spread extensively in the swine industry, affecting both commercial and backyard pigs as well as wild pigs since its introduction into the country in 2019. The control and prevention of ASF greatly depend on the collaborative efforts and cooperation of all stakeholders, veterinarians, policy makers, and other institutions starting from timely reporting of ASF cases to early diagnosis and strict implementation of prevention and control measures. The Philippine government, headed by the Department of Agriculture and Bureau of Animal Industry, has been vigilant in implementing various ASF control and prevention strategies (Figure 4). The implementation of Administrative Circular No. 12, Series of 2019 (National Zoning and Movement Plan for the Prevention and Control of African Swine Fever) was among the initial government responses to the ASF outbreak, designating different zones based on ASF risk levels [59]. The zoning and compartmentalization program of three different regions in the Philippines is based on disease status, protecting unaffected areas while controlling disease in infected areas. Infected zones or areas with confirmed ASF cases are indicated in red color; pink indicates the buffer zones, which are areas adjacent to infected zones and include provinces that still have infected areas but no new ASF cases; yellow indicates areas under surveillance zones, which are high-risk areas due to the volume of pigs, pork, and other animal products for trade and a large number of swine. Light green areas are protected zones, which have no cases of ASF and are considered low-risk areas; green areas are ASF-free areas (Figure 5A).

The Department of Agriculture has also implemented the Administrative Order No. 07 Series of 2021 released by the Department of Agriculture which establishes guidelines for the “Bantay ASF sa Barangay Programme” or BABay ASF Programme, an NASFPCP program that coordinates with the veterinary offices of all provinces to increase surveillance; monitor the movement of live pigs, raw pork, and pork products between provinces or regions; raise awareness of the disease; improve the biosecurity measures of local farms; increase border security for pork products; and enhance recovery and repopulation programs. To effectively implement these, local government units (LGUs) have a list of farms in their area of jurisdiction for easier monitoring and surveillance, and veterinarians are equipped with capacity-building training on how to effectively respond during an ASF outbreak. This includes critical protocols on how to properly collect blood samples from pigs for ASF surveillance and monitoring as well as the proper implementation of stricter border control in each area of jurisdiction. In addition, almost 55% of the provinces in the country have barangay biosecurity officers (BBO) who were also trained to properly implement biosecurity procedures in their area, especially during ASF outbreaks. Checkpoints are currently implemented to monitor the movement of live pigs including processed pork products. A total ban on incoming pigs is being practiced if there is no veterinary health certificate, shipping permit, livestock handlers’ certificate, or transport carrier registration from the government and if pigs came from ASF-infected areas around the country. Prior to the issuance of the above-mentioned permits, an ASF-negative laboratory result from government-accredited laboratories must be presented [38].

For areas reporting an ASF case for the first time, the 1-7-10 Protocol is applied as a rapid response to control the disease. This protocol is regulated by the Department of Agriculture through an administrative order. As in other countries, the Philippines’ main course of action is “stamping-out” ASFV-positive swine herds. In this protocol, all pigs within a 1 km radius circle around the affected farm are to be stamped out within 5 days. On the other hand, stricter disease surveillance is conducted for the detection of other cases in proximity of a 7 km radius, and the movement of pork products is also limited. Within a 10 km radius, surveillance is still conducted, as well as mandatory reporting of pigs exhibiting clinical signs of ASF [29]. In addition, after the infected area has been cleaned and decontaminated, there is a 90-day period during which sentinel animals must not comprise more than 10% of the stocking density [60].

Strengthening the ASF surveillance program and laboratory capacity of the government is also an essential measure for controlling ASF. Veterinarians are engaged in capacity training to better equip them in outbreak investigation and disease diagnosis. Important risk factors in ASF spread, such as infected raw meat, pork products, environmental fomites, vertebrate or invertebrate hosts, contaminated feeds, and wild pigs, should also be monitored and included in the epidemiological model. In a conjoint analysis conducted by Hsu et al., swill or contaminated feed was reported to be the most significant factor influencing ASF spread in the Philippines [59]. Although wild boar or feral pigs are less likely to be associated with the ASF spread, ASF in wild boar may be under-reported in Asia [59,61]. On the other hand, the lack of proper biosecurity in farms may allow stray dogs and cats to enter farm premises and access swine feeds and carcasses, presenting a potential mechanical vector of ASFV. Hsu et al., also reported that commercial swine farmers have stricter biosecurity as they have more resources and capabilities to manage ASF outbreak losses as compared to backyard swine farmers who often rely on government support or compensation during ASF outbreaks [59].

Early ASFV detection in raw pork and processed pork products as well as at a market level is an essential layer of monitoring and disease surveillance required for control of the disease. Disease surveillance in farms alone may not entirely show all the sources of ASFV infection in a given territory, as the virus has proven to be resilient not only in raw pork tissue but in the environment as well, highlighting its longevity in reaching a new host. These possible sources of infection beyond the farm setting may potentially be sources of infection for other farms in different localities, thus leading to a positive feedback loop of ASF cases passing back and forth between ASF-infected zones as long as there is a healthy swine population to exploit. This enhanced approach to disease surveillance will hopefully provide better insight into the complexity of ASF disease control. A better understanding of the disease will hopefully allow policymakers to adapt or improve ongoing ASF disease protocols, regulations, and programs.

Alongside other ASF control programs, the government has also established the Integrated National Swine Production Initiatives for Recovery and Expansion or INSPIRE program, which seeks to recover the hog inventory lost to the ASF outbreak and to help hog raisers repopulate their swine farms. Due to the characteristic high mortality rate and quick spread of the disease, depopulation of an affected swine farm is seen as the most effective method of disease control. However, backyard farms lack the resources and capabilities to manage ASF outbreaks in comparison to commercial farms that have established protocols and resources for depopulation [29,59]. Depopulation is an expensive approach to disease control; therefore, government intervention is most likely needed to compensate farmers, formulate revitalization programs, and coordinate with public health and economic institutions to mitigate the damage of ASF outbreaks. Poor adoption of ASF control and surveillance programs both at the national and local levels is usually reinforced by a lack of incentives. This is another critical aspect of control, without which cooperation may not be guaranteed [62]. This is one of the gaps that can reduce the capacity for action and contribute extensively to the vulnerability of the swine industry against ASF.

Other essential factors in the government-implemented control of ASF are capability and awareness. The African Swine Fever Communication Strategy of the Philippines was created to focus on the immediate communication of information to stakeholders in relation to ASF. It aims to ensure that all aspects relevant to ASF control and prevention are cascaded and disseminated to all stakeholders. The government called it TRACE, and it has identified five major components through which all communications and activities related to ASF will be conducted: (1) understanding the nature, characteristics, behavior, transmission, and epidemiology of ASF; (2) risk and crisis communication, which pertain to the timely reporting of relevant information to the stakeholders, starting from the detection of the disease, investigation, response, and management; (3) Awareness and advocacy which pertain to the development, design, and reproduction of ASF information materials for public awareness; (4) creating networks to strengthen collaborations with various levels of government involved in ASF control; and (5) engagement of partners from industry, academia, non-government organizations, private swine practitioners, and other agencies with ASF concerns. Multi-sectoral efforts and coordination are needed as ASF has implications for food security, safety, and livelihood that affect everyone. The best practices in ASF control and prevention are highlighted and shared among these agencies to implement a synergistic approach to fighting ASF [63]. These actions are all necessary because a lack of effective control and eradication measures can lead to endemization [7].

## 5. ASF Vaccine Trials in the Philippines

Extensive efforts have been focused on ASF vaccine development. Recently, ASFV-G-ΔI177L was developed to be a vaccine candidate that does not revert to its virulent form and has passed the safety tests and obtained regulatory approval [64]. There are two commercial vaccines (NAVET-ASFVAC and AVAC ASF LIVE) developed in Vietnam and have been approved for nationwide use in Vietnam [65]. In the Philippines, vaccination trials have been conducted by the Department of Agriculture (DA) and the Bureau of Animal Industry (BAI) since 2021 [47]. The first two vaccines were from a joint venture between a United States (US) vaccine company and Zoetis and another one is the NAVET-ASFVAC from Vietnam. Both are LAV vaccines, but it was reported that both failed to elicit the desired immune response against ASF [38]. In June 2023, following the positive safety results with a 100% production of antibodies and no indication of clinical signs of ASF in vaccinated pigs or no evidence of viral shedding among 4- to 10-week-old pigs, the certificate of product registration (CPR) for the AVAC ASF LIVE vaccine was recommended by the DA-BAI to the Food and Drug Administration [66]. Furthermore, the Philippine government is currently in the process of procuring ASF vaccines and plans to start widespread immunization by June or July 2024 [66]. With all these developments, reports highlight some challenges that must be resolved to effectively use the vaccines as only younger pigs have been utilized for the vaccine efficacy testing. There is still a need to design or create guidelines for vaccine administration in the Philippines to prevent incorrect vaccine administration. Hence, depopulation of or culling infected pigs is still the only effective prevention and control strategy being implemented in affected areas in the Philippines.

## 6. Conclusions and Future Directions

This review provides an overview of the epidemiology, surveillance, control, and prevention strategies for ASF in the Philippines. Geographically, as an archipelago, it has its advantages and disadvantages with regard to the risk of disease transmission. Strict implementation in ports of entry on the islands can prevent the entry risk of disease; however, as the islands are scattered, logistics for support in infected locations is very challenging. The implementation of zoning and compartmentalization of the areas based on their ASF risk has been instrumental in reducing active cases in the country. With a thorough evaluation of the different measures implemented during ASF outbreaks, a multi-sector collaborative approach is evidently important in combating and containing the spread of ASF. Collaboration between the government, stakeholders, academic/research institutions, private sector, and local government units is necessary to ensure the successful implementation of the ASF control and prevention strategies currently implemented in the country. This paper also emphasizes the significance of early diagnosis with improved surveillance, advocating good biosecurity measures including quarantine protocols, movement control, and immediate response for the successful control and prevention of ASF introduction and spread. Moreover, continuous research and development efforts are needed such as genetic characterization of the locally isolated ASFV and risk transmission analysis to further enrich our knowledge of the nature of ASFV and to implement science-based control and prevention strategies.

## Figures and Tables

**Figure 1 animals-14-01816-f001:**
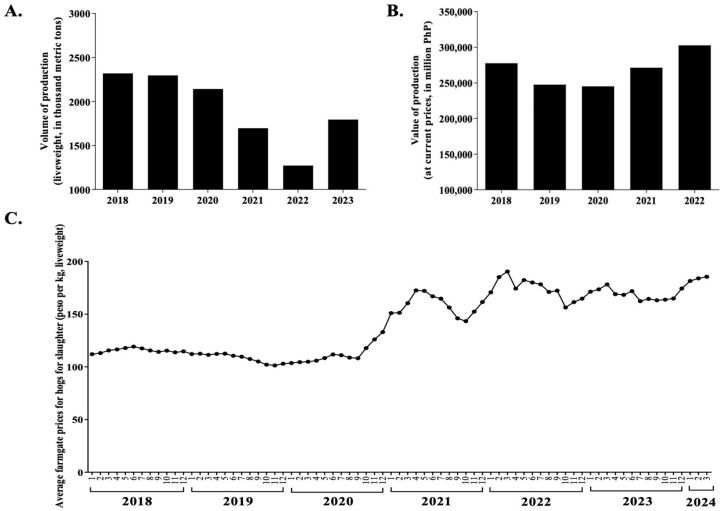
Overview of the swine industry in the Philippines from 2018 to 2024. (**A**) Volume of hog production in the Philippines. Values are expressed as live weight and represent the volume of locally raised hogs disposed for slaughter. (**B**) Value of hog production per year expressed in Philippine currency. At the time of writing, no data are available for the value of production for 2023 and 2024 current prices. (**C**) Monthly average of farmgate price received by raisers. All data and values presented were retrieved and compiled from the Special Release of Swine Situation Report by the Philippines Statistics Authority (PSA). PHP, Philippine peso; 1, January; 2, February; 3, March; 4, April; 5, May; 6, June; 7, July; 8, August; 9, September; 10, October; 11, November; 12, December.

**Figure 2 animals-14-01816-f002:**
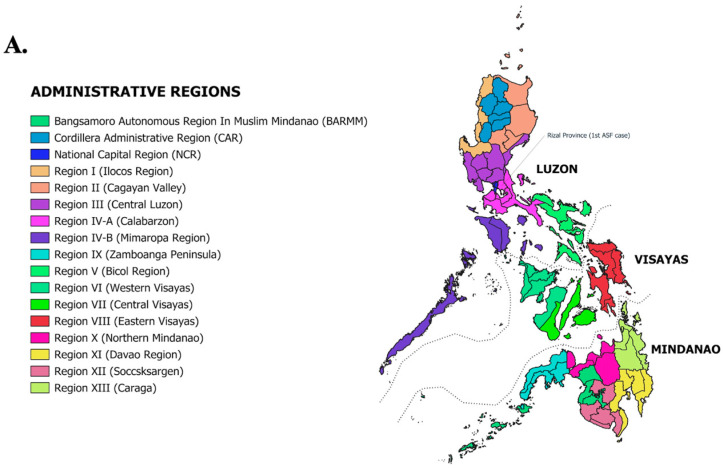
Distribution of swine inventory per region from 2019 to 2022. (**A**) Map of the Philippines with its administrative regions. Map was created in QGIS version 3.30. (**B**) Values presented represent the total number of swine heads including small hold, semi-commercial, and commercial farms from each region at the time of report dates indicated. All data and values presented were retrieved from the Special Release of Swine Situation Report by the Philippines Statistics Authority (PSA) from their reports in 2019 (1 July 2019), 2020 (20 June 2020), 2021 (30 September 2021), 2022, 2023 and first quarter of 2024.

**Figure 3 animals-14-01816-f003:**
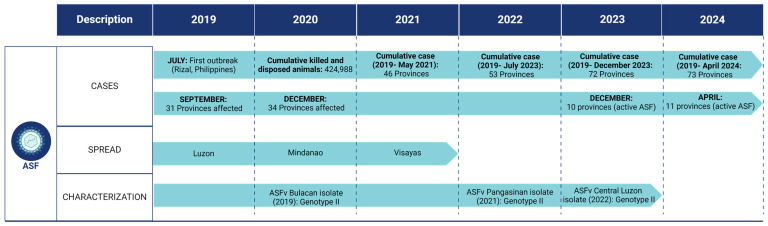
Timeline of ASF outbreaks and epidemiology in the Philippines. ASF, African swine fever; ASFv, ASF virus. The figure was created in BioRender.com (accessed on 15 March 2024).

**Figure 4 animals-14-01816-f004:**
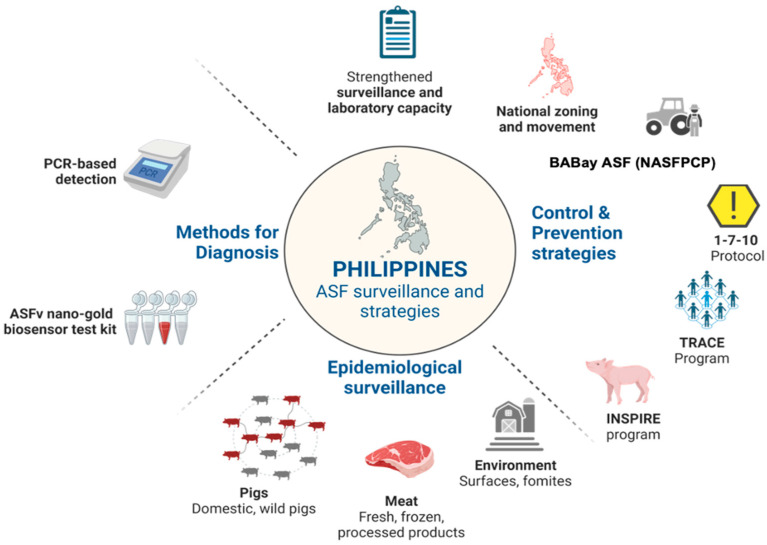
Summary of the Philippine Government’s ASF Surveillance and Prevention and Control Programme Strategies. ASF, African swine fever; BABay ASF Program, Bantay ASF Sa Barangay Program; INSPIRE, Implementation of the Integrated National Swine Production Initiatives for Recovery and Expansion; NTFAD, National Task Force on Animal-Borne Diseases. Figure was created in Biorender.

**Figure 5 animals-14-01816-f005:**
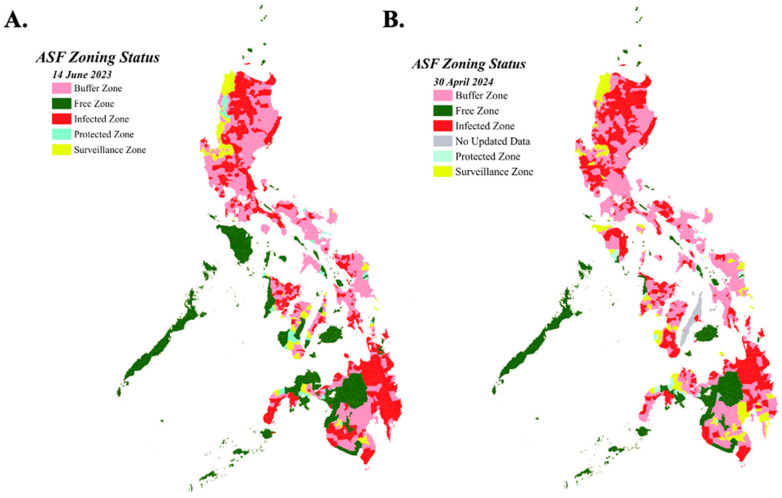
ASF zoning map of cases in the Philippines as of 14 June 2023 (**A**) and 30 April 2024 (**B**). Maps show the zoning and compartmentalization status of the different municipalities based on ASF disease risk. Data presented were retrieved from the update on zoning status of regions in relation to the implementation of the national zoning and movement plan for African Swine Fever (ASF) by the Bureau of Animal Industry, Philippines with permission. Map was created in QGIS version 3.30.

**Figure 6 animals-14-01816-f006:**
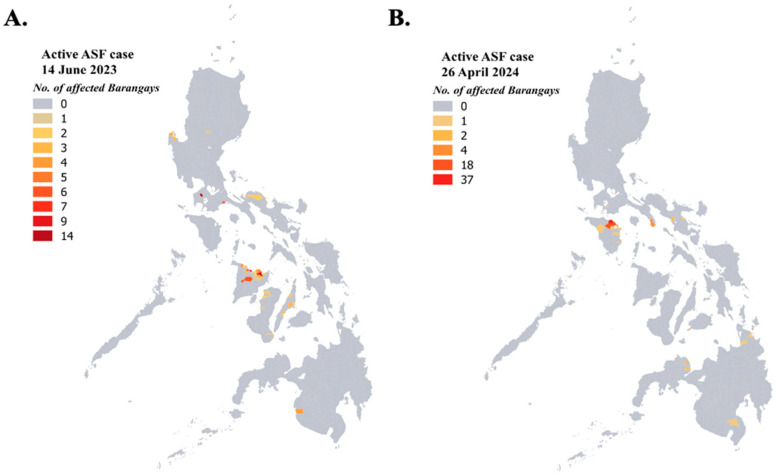
Geographical distribution of ASF active cases in the Philippines as of 14 June 2023 (**A**) and as of 26 April 2024 (**B**). Maps show reported location of active cases of ASF from different municipalities. Colors represent the number of affected barangays (smallest political unit) with ASF active cases for each municipality. Active cases refer to the current or ongoing confirmed presence of ASFV in the area. Data presented were retrieved from the ASF updates by the Bureau of Animal Industry, Philippines with permission. Map was created in QGIS version 3.30.

## Data Availability

All the data presented in this study are available in this article.

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
