# Peer review of "African Swine Fever in the Philippines: A Review on Surveillance, Prevention, and Control Strategies"

_animals, 2024, doi:10.3390/ani14121816_

Round 1

Reviewer 1 Report

Comments and Suggestions for Authors

The article describes the epidemiological situation regarding the spread of the ASF virus in the Philippines. The topic of the work is very current and important as the ASF virus still poses a serious threat to pig farming in many places around the world. However, the work requires careful linguistic correction and improvement of the organization of the article. I also suggest adding some information.

Detailed comments:

Abstract: Please provide the info on what is the purpose of the article and what will be the content

line 52 ASF vaccines are now being produced by some companies and the works on the development of a new vaccines are ongoing - please add some info on current status of ASF vaccine development around the world

line 131 Montgomery?

line 214 What are the advantages of a new kit ? is it cheaper? what about the sensitivity? Is there any scientific paper describing the validation of this kit?

line 241-243Please rewrite the sentence - it is unclear. What are the current regulations regarding pork import in Philipiness and neighbouring countries?

line 253 I am not sure if the import has such a big impact if the outbreak is already in the country.

line 273 which country?

Figure 6 please define active cases

line 337 completed?

line 343 sentinel?

Comments on the Quality of English Language

English is difficult to understand.

Author Response

[Animals] Manuscript ID: animals-3017127 

We acknowledge and appreciate the reviewer’s insightful comments and thank the reviewers for providing us with a very thorough review of our manuscript and recommendations for improvement. Your constructive feedback is invaluable to enhance the quality of our manuscript. Below are our point-by-point responses. We really hope these will meet with your approval.

Reviewer 1

Comments and Suggestions for Authors

The article describes the epidemiological situation regarding the spread of the ASF virus in the Philippines. The topic of the work is very current and important as the ASF virus still poses a serious threat to pig farming in many places around the world. However, the work requires careful linguistic correction and improvement of the organization of the article. I also suggest adding some information.

Detailed comments:

  1. Abstract: Please provide the info on what is the purpose of the article and what will be the content

Response: The following sentences have been added in the abstract (Lines 32-37) in the revised manuscript: “It is worth mentioning that the government’s efforts toward a comprehensive ASF surveillance and epidemiological investigation into the possible ASFV sources or transmission pathways are the most important measures in the prevention and control of ASF outbreaks. This review article provides a comprehensive overview of the current swine industry and ASF situation in the Philippines which includes its epidemiology, surveillance, prevention and control strategies.”

  1. Line 52 ASF vaccines are now being produced by some companies and the works on the development of a new vaccines are ongoing - please add some info on current status of ASF vaccine development around the world

Response: The following sentences have been added in the introduction (Lines 52-59) in the revised manuscript: “The ASFV genome is very large and complex which contributes to the difficulty of developing an effective vaccine. However, extensive efforts around the world have shown promising progress directed toward effective vaccine development. Various approaches have been employed which include inactivated, subunit, DNA, virus-vectored and live-attenuated (LAV) ASF vaccines [11]. Among these approaches, inactivated and subunit vaccines are safe but do not induce protective immunity [12-14] while live-attenuated (LAV) ASF vaccines are the most promising which exhibit wide range of safety and efficacy against ASF [15].” In addition, a new topic on “ASF vaccine trials in the Philippines” was also added in the revised manuscript (Lines 460-483).

  1. line 131 Montgomery?

Response: The words “by Montgomery” in Line 143 was deleted in the revised manuscript.

  1. line 214 What are the advantages of a new kit ? is it cheaper? what about the sensitivity? Is there any scientific paper describing the validation of this kit?

Response: The following sentences were added in the revised manuscript (Lines 241-246) as additional information on the locally developed test kit: “This test kit was validated in 41 farms (32 commercial, 9 backyard) in 7 provinces with 90-100% sensitivity and 77-85.7% specificity in parallel with RT-PCR [46]. Furthermore, it can be used in various samples such as oral and rectal swabs, feces, water, semen, feeds, blood, environmental swabs and raw and processed meat samples to detect the P72 gene of ASFV. It can be used as a screening test which is on standby at the port of entries for meat products entering into the country [47].”

As for the cost, yes it is definitely cheaper, around (Php 400.00 or USD 7.00 per sample) [46].

  1. line 241-243 Please rewrite the sentence - it is unclear. What are the current regulations regarding pork import in Philipiness and neighbouring countries?

Response: The following sentence was added (Lines 268-271) in the revised manuscript: “Despite imposed regulations and import taxes designed in each country to control the entry of pork and processed products, illegal importation remains a significant concern on the control and prevention of ASF outbreaks. ”  In addition, the sentence in Lines 241-243 (old version) was rewritten as suggested (Lines 273-276-revised): “On the other hand, Japan also evaluated the risk of ASFV transmission from pork products that brought by tourists. The findings indicated that the likelihood of ASFV entering Japan increases substantially as more of these pork products are being fed to pigs in Japan [50].”

  1. line 253 I am not sure if the import has such a big impact if the outbreak is already in the country.

Response: The sentences in Lines 282-287 were rewritten in the revised manuscript (Lines 282-287): “The risk of ASF transmission among domestic pigs into other areas that are still ASF-free may be intensified with ASFV-contaminated raw meat and processed pork products sourced locally and imported from other countries. By imposing strict sanitary and import restrictions on raw meat and processed products, the entry of other ASFV variants or recombinants can be prevented, thus minimizing the impact of new outbreaks.”

  1. line 273 which country?

Response: Line 273 (old version) was rewritten to “The authors reported that there was a much lower ratio of ASF in wild boars to farm outbreaks in Asia as compared to Europe. Aside from South Korea with 605 infected wild boar carcasses since 2020, only China and Laos have reported wild boar infections with 23 cases [54].” In the revised manuscript (Lines 307-310).

  1. Figure 6 please define active cases

Response: The word active cases was clearly defined in Figure 6 legend as suggested by the reviewer. Hence, the sentence, “Active cases refer to the current or ongoing confirmed presence of ASFV in the area.” was added in lines 335-336 of the revised manuscript.

  1. line 337 completed?

Response: The word “completed” in line 337 was changed to “to be stamped out” in the revised manuscript (Line 390).

  1. line 343 sentinel?

Response: “Sentinel” animals are specific pigs placed in a previously ASF-infected farm/area which will serve as animal monitors for ASF infection prior to repopulation of the area/farm. They serve as early warning system for any resurgene of ASF infection. Moreover, the use of sentinel pigs contributes to the validation of the efficacy of biosecurity protocols such as cleaning and disinfection before the reintroduction of a larger herd into the farm.

  1. Comments on the Quality of English Language: English is difficult to understand.

Response: We appreciate the reviewer’s comments and suggestions. The revised manuscript undergone English editing. 

Reviewer 2 Report

Comments and Suggestions for Authors

Dear P.T. Authors,

The manuscript is well-written and presents significant information concerning ASF surveillance, prevention, and control strategies.

I would like to emphasize that I highly appreciate the figures included in the article, as they significantly aid in understanding the text.

The current version of this article requires almost no corrections; however, due to my duty as a reviewer, I would like to propose some suggestions to further improve the quality of this article. Please find my remarks below:

  1. Lines 92 and 94: You mention “farm gate price” and “farmgate price” – please standardize the spelling.
  2. Figure 1 c and d present, respectively: inventory of the actual number of swine in heads and volume of swine slaughtered in heads. According to the presented numbers, the number of slaughtered pigs exceeds the inventory number of swine heads. Please explain in the main text what this discrepancy stems from and identify the main pork importers for the Philippines. This information is important and could be used further in the discussion, indicating possible risk sources.

Author Response

[Animals] Manuscript ID: animals-3017127 

We acknowledge and appreciate the reviewer’s insightful comments and thank the reviewers for providing us with a very thorough review of our manuscript and recommendations for improvement. Your constructive feedback is invaluable to enhance the quality of our manuscript. Below are our point-by-point responses. We really hope these will meet with your approval.

Reviewer 2

Comments and Suggestions for Authors

The manuscript is well-written and presents significant information concerning ASF surveillance, prevention, and control strategies. I would like to emphasize that I highly appreciate the figures included in the article, as they significantly aid in understanding the text.

The current version of this article requires almost no corrections; however, due to my duty as a reviewer, I would like to propose some suggestions to further improve the quality of this article. Please find my remarks below:

  1. Lines 92 and 94: You mention “farm gate price” and “farmgate price” – please standardize the spelling.

Response: The phrase “farm gate price” has been changed to “farmgate price” in the revised manuscript as suggested by the reviewer.

2. Figure 1 c and d present, respectively: inventory of the actual number of swine in heads and volume of swine slaughtered in heads. According to the presented numbers, the number of slaughtered pigs exceeds the inventory number of swine heads. Please explain in the main text what this discrepancy stems from and identify the main pork importers for the Philippines. This information is important and could be used further in the discussion, indicating possible risk sources.

Response: Figure 1 has been updated with new data which includes the data of the first quarter (January 2024-April 2024) of year 2024 as reflected in Figure 1C in the revised manuscript. Moreover, after careful analysis, we have deleted Figure 1C (inventory of number of swine heads) and Figure 1D (volume of slaughtered pigs) to streamline the presentation of the data, and the figure on the inventory number of swine heads was removed due to duplication in Figure 2B where data on the total number of swine heads were presented per region.

Reviewer 3 Report

Comments and Suggestions for Authors

The topic of the review is interesting but the content does not correspond to what is reported in the title. The method applied for the review is missing. In some cases the terminology is scientifically not adequate (“ASF proliferation”, “technical element of ASF”) and the writing also needs to be improved.

Reference is made to the epidemiology of the disease in the Philippines but there are no references to how the disease may have been introduced into the country while several information are reported about the disease in other countries that should not be subject to the review.

The focus should be on surveillance and control of the disease but the main reference is to laboratory diagnosis, ..models but there is no reference to the surveillance activities to be carried out in the field, do veterinarians visit farms? (regardless of the outbreaks) is there a census of farms, is it reliable, are the movements of the animals under control...?

Laboratory diagnostics is only one element of surveillance, and if everything that comes before it doesn't work, laboratory diagnosis is also fatally weakened.

For example, early detection is certainly important but if there are no veterinarians in the field capable of raising suspicions and restricting the suspected herd, controlling or blocking the movement of animals, laboratory diagnosis is of little use, if the focus is “disease control, surveillance, prevention”, as reported in the title of the manuscript.

Not clear what is difference between small and commercial farms, are they subject to the same control measures?

Worth mentioning that the eradication of a disease depends on the combination of several control actions, all of which are equally important.

Comments on the Quality of English Language

English should be improved as well as writing. The terminology, in epidemiological terms, is not always adequate.

Author Response

[Animals] Manuscript ID: animals-3017127 

We acknowledge and appreciate the reviewer’s insightful comments and thank the reviewers for providing us with a very thorough review of our manuscript and recommendations for improvement. Your constructive feedback is invaluable to enhance the quality of our manuscript. Below are our point-by-point responses. We really hope these will meet with your approval.

Reviewer 3

Comments and Suggestions for Authors

  1. The topic of the review is interesting but the content does not correspond to what is reported in the title. The method applied for the review is missing. In some cases the terminology is scientifically not adequate (“ASF proliferation”, “technical element of ASF”) and the writing also needs to be improved.

Response: The phrase “technical element of ASF” has been changed to “understanding the nature and characterisitcs, and behavior, transmission and epidemiology of ASF…” in Lines 446-447. Upon checking, there is no “ASF proliferation” phrase in the revised manuscript. We also sent the manuscript for English editing as suggested by the reviewer.

  1. Reference is made to the epidemiology of the disease in the Philippines but there are no references to how the disease may have been introduced into the country while several information are reported about the disease in other countries that should not be subject to the review.

Response: The following sentences were added in Lines 152-159 in the revised manuscript: “According to the retrospective study on the epidemiology of ASF outbreaks in the Philippines, the first case of ASF in Rodriguez, Rizal was attributed to swill feeding practices of hog raisers in the area [38]. The question on how the disease may have been introduced into the Philippines remains unanswered due to lack of concrete evidence and only causal inference as to the source of infection [39]. However, in June 2019 prior to the detection of ASFV in pigs, it was reported that a luncheon meat seized at an international airport was positive of ASFV as confirmed by PCR performed at a Regional Animal Disease Diagnostic Laboratory (RADDL) [40].“

            With regards to the several information about ASF in other countries, we wanted to include them in the paper to justify that ASF surveillance should not only focus on its detection in live pigs but also take into consideration other risk factors such as raw meat and other processed pork products entering into the country especially from international travellers coming from countries with ASF outbreaks.

  1. The focus should be on surveillance and control of the disease but the main reference is to laboratory diagnosis, ..models but there is no reference to the surveillance activities to be carried out in the field, do veterinarians visit farms? (regardless of the outbreaks) is there a census of farms, is it reliable, are the movements of the animals under control...?

Response: The following sentences were added in the revised manuscript (Lines 346-352): “

“The control and prevention of ASF greatly depend on the collaborative efforts and cooperation of all stakeholders, veterinarians, policy makers and other institutions starting from timely reporting of ASF cases to early diagnosis and strict implementation of prevention and control measures. The Philippine government, headed by the Department of Agriculture and Bureau of Animal Industry, have been vigilant in implementing various ASF control and prevention strategies (Figure 4).”

            Other sentences were also added in Lines 372-385 in the revised manuscript:

“To effectively implement these, local government units (LGUs) have list of farms in their area of jurisdiction for easier monitoring and surveillance and veterinarians were equipped with capacity-building trainings on how to effectively respond during an ASF outbreak. This includes critical protocols on how to properly collect blood samples from pigs for ASF surveillance and monitoring as well as the proper implementation of stricter border control in each area of jurisdiction. In addition, almost 55% of the provinces in the country have barangay biosecurity officers (BBO) who were also trained to properly implement biosecurity procedures in their area especially during ASF outbreaks. Check points are currently implemented to monitor movement of live pigs including processed pork products. Total ban of incoming pigs is being practiced if there is no veterinary health certificate, shipping permit, livestock handlers’ certificate and transport carrier registration from the government and if pigs came from ASF-infected areas around the country. Prior to issuance of the above-mentioned permits, an ASF-negative laboratory result from government-accredited laboratories must be presented [38].”

  1. Laboratory diagnostics is only one element of surveillance, and if everything that comes before it doesn't work, laboratory diagnosis is also fatally weakened. For example, early detection is certainly important but if there are no veterinarians in the field capable of raising suspicions and restricting the suspected herd, controlling or blocking the movement of animals, laboratory diagnosis is of little use, if the focus is “disease control, surveillance, prevention”, as reported in the title of the manuscript.

Response: The paragraph (Lines 372-385) was added in the revised manuscript as mentioned above (Response to Comment #3). In addition, the following sentences were added in Lines 406-411 in the revised manuscript indicating the importance of biosecurity in the farms in controlling ASF outbreaks: “On the other hand, the lack of proper biosecurity in farms may allow stray dogs and cats to enter farm premises and access swine feeds and carcasses, presenting a potential mechanical vector of ASFV. Hsu et al., also reported that commercial swine farmers have stricter biosecurity as they have more resources and capabilities to manage ASF outbreak losses as compared to backyard swine farmers who often rely on government support or compensation during ASF outbreaks [59].”

  1. Not clear what is difference between small and commercial farms, are they subject to the same control measures?

Response: Small-hold or backyard farms and commercial farms are the types farms based on hog population in the Philippines. According to PSA (2021), a backyard or smallhold farm can be distinguished from a commercial farm by the quantity of heads the producer raises. Backyard/smallhold farms are those that raise one or more of the following types of livestock: 1–20 finishers without piglets, 1–40 piglets, or 1–10 sows with 1–21 piglets. On the other hand, a commercial farm raises more than 10 sows with 22 piglets, 41 or more piglets, or 21 finishers or more.

Yes, all swine farms, smallhold/backyard or commercial, must follow the regulatory procedures of the government not only in terms of ASF control but also of other equally important swine diseases.

  1. Worth mentioning that the eradication of a disease depends on the combination of several control actions, all of which are equally important.

Response: We agree to the reviewer’s comments that there should be a multi-sectorial approach from the different agencies (government or private) in combating ASF.

  1. Comments on the Quality of English Language

English should be improved as well as writing. The terminology, in epidemiological terms, is not always adequate.

Response: We appreciate the reviewer’s comments and suggestions. The revised manuscript undergone English editing.

Reviewer 4 Report

Comments and Suggestions for Authors

In this Review, Fernandez-Colorado and colleagues present an overview of the epidemiology, surveillance, control, and prevention strategies for ASF in the Philippines. This paper underscores the importance of early diagnosis, enhanced surveillance, and robust biosecurity measures, such as quarantine protocols, movement control, and immediate response, for the successful control and prevention of ASF introduction and spread. Overall, the article has a clear structure and provides a thorough and complete analysis. Nevertheless, there are some major issues that require clarification.

Major issues:

1. The introduction needs to include new features of ASFV, such as changes in virulence and genetic recombination. This will also be a focus of future monitoring in the Philippines.

2. Please summarize the molecular epidemiology of ASFV that is missing in Part 3. Analysis and discussion of whether there have been variations, especially in the process of ASFV prevalence.

3. Please briefly introduce the research progress of the Philippines on ASF vaccine.

4. Please reflect the overall situation of ASF prevalence and losses in the Philippines from 2019 to 2024 in the abstract.

5. Are there new experiences and measures in the Philippines that are worth drawing on compared to other countries in the prevention and control of ASF?

Author Response

[Animals] Manuscript ID: animals-3017127 

We acknowledge and appreciate the reviewer’s insightful comments and thank the reviewers for providing us with a very thorough review of our manuscript and recommendations for improvement. Your constructive feedback is invaluable to enhance the quality of our manuscript. Below are our point-by-point responses. We really hope these will meet with your approval.

Reviewer 4

Comments and Suggestions for Authors

In this Review, Fernandez-Colorado and colleagues present an overview of the epidemiology, surveillance, control, and prevention strategies for ASF in the Philippines. This paper underscores the importance of early diagnosis, enhanced surveillance, and robust biosecurity measures, such as quarantine protocols, movement control, and immediate response, for the successful control and prevention of ASF introduction and spread. Overall, the article has a clear structure and provides a thorough and complete analysis. Nevertheless, there are some major issues that require clarification.

Major issues:

  1. The introduction needs to include new features of ASFV, such as changes in virulence and genetic recombination. This will also be a focus of future monitoring in the Philippines.

Response: The following sentences were added in the revised manuscript (Lines 70-77):

“…with ASFV genotype II as the prevailing strain. Currently, there are 24 known ASFV genotypes, however, a significant new information on the genetic features of ASFV in relation to changes in virulence and genetic recombination, has been reported in China [18]. In this study, three recombinants of genotypes I and II were detected in pigs and based on the B646L gene, these recombinants are categorized as genotype I due to their genetic similarity. However, 56% of their genomes are derived from genotype II ASFV. Furthermore, one of the recombinant viruses has been shown to be highly lethal and transmissible in pigs in animal experiments [18].”

  1. Please summarize the molecular epidemiology of ASFV that is missing in Part 3.Analysis and discussion of whether there have been variations, especially in the process of ASFV prevalence.

Response: The following sentence was added in the revised manuscript (Lines 199-202). “As of writing this review, 20 Philippine ASFV isolate whole genome sequences are available in NCBI GenBank; however, reports on these isolates' genetic characterization and analysis including genetic variation still need to be elucidated and is beyond the scope of this review.” 

  1. Please briefly introduce the research progress of the Philippines on ASF vaccine.

Response: The following senetences have been added to the introduction (Lines 52-59) in the revised manuscript: “The ASFV genome is very large and complex which contributes to the difficulty of developing an effective vaccine. However, extensive efforts around the world have shown promising progress directed toward effective vaccine development. Various approaches have been employed which include inactivated, subunit, DNA, virus-vectored and live-attenuated (LAV) ASF vaccines [11]. Among these approaches, inactivated and subunit vaccines are safe but do not induce protective immunity [12-14] while live-attenuated (LAV) ASF vaccines are the most promising which exhibit wide range of safety and efficacy against ASF [15].” In addition, a new topic on “ASF vaccine trials in the Philippines” was also added in the revised manuscript (Lines 460-483).

  1. Please reflect the overall situation of ASF prevalence and losses in the Philippines from 2019 to 2024 in the abstract.

Response: The following senctence was added in the introduction instead of in the abstract due to limited number of words required. In Lines 85-88, the following sentences were added: “Since its detection in 2019, 89% (73 out of 82) of the provinces were already affected with 5 million pigs were already killed according to the Pork Producers Federation of the Philippines which resulted in approximately PhP 200 billion or more losses as [23].”

  1. Are there new experiences and measures in the Philippines that are worth drawing on compared to other countries in the prevention and control of ASF?

Response: The Philippines is an archipelago comprising approximately 7,641 islands, geographically grouped into three main island groups: Luzon, Visayas, and Mindanao. The country is further divided into 17 regions, 82 provinces, and 42,029 barangays.

The following sentences were added in the conclusion and future directions in the revised manuscript (Lines 486-491): “Geographically, as an archipelago, it has its advantages and disadvantages with regards to the risk of disease transmission. Strict implementation in ports of entry on the islands can prevent the entry risk of disease; however, as the islands are scattered, logistics for support in infected locations is very challenging. The implementation of zoning and compartmentalization of the areas based on their ASF risk has been instrumental in reducing active cases in the country.”

Round 2

Reviewer 1 Report

Comments and Suggestions for Authors

All my questions have been answered